# Response of Sediment Dynamics to Tropical Cyclones under Various Scenarios in the Jiangsu Coast

Can Wang [1], Chengyi Zhao [1,*], Gang Yang [2], Chunhui Li [2], Jianting Zhu [3] and Xiaofei Ma [1]

1   School of Geographical Sciences, Nanjing University of Information Science and Technology,
    No. 219 Ningliu Road, Nanjing 210044, China; 20211211008@nuist.edu.cn (C.W.)
2   School of Marine Sciences, Nanjing University of Information Science and Technology, Nanjing 210044, China
3   Department of Civil and Architectural Engineering, University of Wyoming, Laramie, WY 82071, USA;
    jzhu5@uwyo.edu
*   Correspondence: zhaocy@nuist.edu.cn; Tel./Fax: +86-25-58235199

**Abstract:** The Jiangsu Coast (JC), China, is an area susceptible to the impact of tropical cyclones (TCs). However, due to the lack of available on-site observation data, nearshore sedimentary dynamic processes under the impact of TCs have not been fully explored. This study developed a 3D wave–current–sediment numerical model for the JC based on the Finite Volume Community Ocean Model (FVCOM) to investigate sediment dynamic responses to TCs under various scenarios, including different tracks, intensities of TCs and tidal conditions. The validation results demonstrated the model's satisfactory performance. According to the simulation results, typhoons can significantly impact the hydrodynamics and sediment dynamics. During Typhoon Lekima in 2019, strong southeasterly winds substantially increased the current velocity, bottom stress, wave height, and suspended sediment concentration (SSC). Three typical landfall-type typhoons, with prevailing southeasterly winds, brought significant sediment flux from southeast to northwest along the coast, while the typhoon that moved northward in the Yellow Sea induced a relatively small sediment flux from north to south. Typhoons could also induce stripe-like erosion and deposition, which is closely related to seafloor topography, resulting in seabed thickness variations of up to ±0.3 m. Additionally, strengthening typhoon wind fields can lead to increased sediment flux and seabed morphological changes. Typhoon Winnie, particularly at spring tide, had a greater impact on sediment dynamics compared to other landfall typhoons. Numerical simulations showed that the typhoon-induced net sediment flux within the spring tidal cycle could increase by 80% to 100% compared to the neap tidal cycle, indicating the significant influence of tidal conditions on sediment transport during TC events.

**Keywords:** Jiangsu Coast; numerical modeling; tropical cyclone; sedimentary dynamic processes; morphological change

## 1. Introduction

Tropical cyclones (TCs) are extreme weather systems, often accompanied by gales, rainstorms, huge waves, and storm surges [1]. The strong interaction between the atmosphere and the ocean can significantly affect coastal hydrodynamics, and the seabed morphology may undergo significant changes after a series of processes such as sediment resuspension, transport, and sedimentation [2]. In some cases, TC events may contribute nearly half of the seasonal net sediment transport [3], even two orders of magnitude higher than that under fair weather conditions [4,5]. Theoretical and modeling results indicate that global warming may lead to an increasing potential for TCs destruction, which may threaten the safety of residents' lives and properties in coastal areas [6]. Therefore, it is essential to enhance our comprehension of sediment dynamics under TC effects to provide a theoretical basis for coastal management and disaster prevention and mitigation.

According to on-site observations, both Typhoons No. 8114 and No. 9711 significantly increased the water level and wave height along the eastern coast of China, causing

significant changes in the morphology of tidal flats [7,8]. Yang et al. [1] found that the strong wave current interaction caused by Typhoon No. 1909 greatly increased bottom stress that determined sediment transport. Zhang et al. [9] found that typhoons directly landing along the coast of Jiangsu increased storm surges in the radial sand ridge (RSR) area by up to nearly 3 m. The waves induced by storms dissipated wave energy in shallow waters and significantly enhanced the process of sediment resuspension, thereby affecting sedimentation dynamics [10]. Ma et al. [2] found that alongshore currents played a crucial role in coastal morphology changes during a storm. Xia et al. [11] found that typhoons that made landfall on the south side of the Yangtze River Estuary experienced greater storm surges than those on the front and north sides. When the typhoon storm surge, upstream flood peak and astronomical spring tide are combined, the storm surge is more significant. Similarly, extreme storm surges may induce coastal inundation in Yangtze Estuary regions, and the relative sea-level rise could reach 0.7 m in the worst-case scenario [8].

Jiangsu Province is located on the eastern coast of China, which is often affected by active tropical cyclones from the northwest Pacific. There is an average of three typhoons per year affecting Jiangsu Province [9]. Typhoons that threaten the Jiangsu Coast (JC) can be divided into three types: typhoons that move northward after landfall in Zhejiang or Fujian Province, typhoons that directly land along the coast of Jiangsu and typhoons that operate in the coastal areas. These types of typhoons have a significant impact on the hydrodynamics and sediment dynamics in the JC. While several studies have investigated the response of JC tidal flats to TC events, they either only considered the impact of specific typhoons or neglected the differences in the impact of typhoons on coastal sediment dynamics under different tidal conditions [2,7,12]. Therefore, the sedimentary dynamic response to TCs under various future scenarios—such as different tracks, intensities of TCs and different tidal conditions remains unclear. Furthermore, the adverse weather and oceanic conditions experienced during typhoons have posed significant obstacles to direct observation efforts, bringing challenges in using calibrated models to reconstruct the impacts of historical TCs on the sediment dynamics [10,13,14].

In this study, we developed a coupled wave–current–sediment model for the JC and explored the hydrodynamics, sediment transport, and morphological changes during typhoon events. The main objectives were to: (1) investigate the response of hydrodynamics and sediment dynamics to TC events, (2) compare the effects that different tracks and intensities of typhoons have on the hydrodynamics and sediment dynamics, and (3) examine the tidal effects on the sediment dynamics during TC events. To achieve these objectives, we adopted the Finite Volume Community Ocean Model (FVCOM) and SWAN model as the ocean model and wave model, respectively. Subsequently, a total of eleven sets of experiments were designed based on different typhoon tracks, intensities, and tidal conditions.

## 2. Regional Setting and Study Domain

The Jiangsu Coast (JC) is located on the west coast of the South Yellow Sea and is an important eastern coastal area in China. The coastline starts from the Xiuzhen River's mouth at the border of Jiangsu and Shandong in the north and ends at the northern mouth of the Yangtze River in the south. The mainland coastline of the JC is 734.84 km long, and the length of the island coastline is about 84.74 km. This study took the entire coastal area of Jiangsu Province as the study domain, with a latitude and longitude range of 31.5°~35.1° N and 119°~123° E [15]. The hydrodynamics in this region is primarily governed by semidiurnal tides, with large tidal range that can reach up to 7 m.

The offshore shoals cover a vast area of the JC, with a total area of 5000 km$^2$ of uncultivated coastal mudflats. According to sediment distribution characteristics, the JC can be roughly divided into four sections from north to south. The northern part is the sandy, bedrock, and muddy coast of Haizhou Bay, the central part is the eroded coast of the abandoned Yellow River Delta, the central southern part is the radial sand ridge area, and the southern part is the coast of the Yangtze River Delta [15]. In general, the coastline is

mainly muddy, except in the north. Between 1128 and 1855, the ancient Yellow River carried a large amount of sediment into the Yellow Sea, causing sedimentation and significant eastward expansion of the JC. After the Yellow River returned to the Yellow Sea in 1855, the coastal dynamic conditions had been changed [16]. After erosion, transportation, and sedimentation, the typical muddy nearshore area took its current shape. The RSR system in the South Yellow Sea is a unique landform that extends from the Yangtze River Delta to the Sheyang River north of Jianggang, spanning 200 km and 90 km in length. Over 70 sand ridges extend from the Jianggang offshore to the Yellow Sea, which is considered to be one of the most remarkable coastal landscapes in the world [1,17,18].

The JC is located in a transitional monsoon climate region between the northern subtropical and warm temperate zones and is influenced by both oceanic and continental climates. The climate is distinct in four seasons, cold in winter and hot in summer, with an annual mean temperature of 13–16 °C and average annual precipitation of 804–1150 mm, mainly from June to September. The annual mean offshore wind speed in Jiangsu is relatively high, reaching 6–10 m/s. In winter, the prevailing wind direction is northeasterly, while in summer it shifts to southeasterly. The frequency of occurrence of significant offshore wave heights greater than 2 m in Jiangsu is relatively low, with the highest significant wave heights in Rudong's outer sea and Haizhou Bay, while the lowest are in the abandoned Yellow River Estuary. The JC is susceptible to TCs in summer between May and October, peaking in August and September. From 2000 to 2008, the southern region of the JC experienced an average of 1.86 typhoons per year, while the northern part was impacted by an average of 0.72 typhoons annually.

In this study, we considered five typical typhoons affecting the JC since 1997 (Figure 1a). Typhoon Lekima in 2019 was a typical strong TC that affected the JC. It formed in the northwest Pacific on 4 August 2019, made landfall in Zhejiang Province on 10 August with maximum wind speeds (Vmax) reaching 52 m/s, and then moved towards Jiangsu Province with reduced wind speed (Vmax = 23 m/s). The typhoon eventually dissipated over the Bohai Sea on 13 August 2019. Its passage resulted in significant economic losses in China, exceeding CNY 74 billion [1,19,20]. As there were some field observation data available during Typhoon Lekima, this study took this typhoon as an example to investigate the dynamic processes on the JC under the impact of TCs.

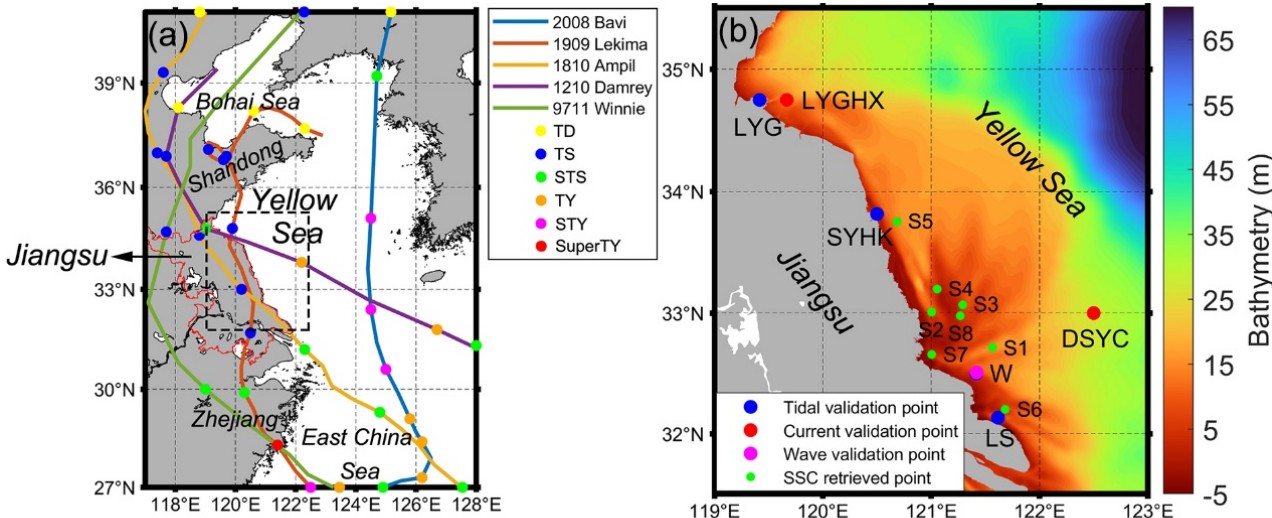

**Figure 1.** *Cont.*

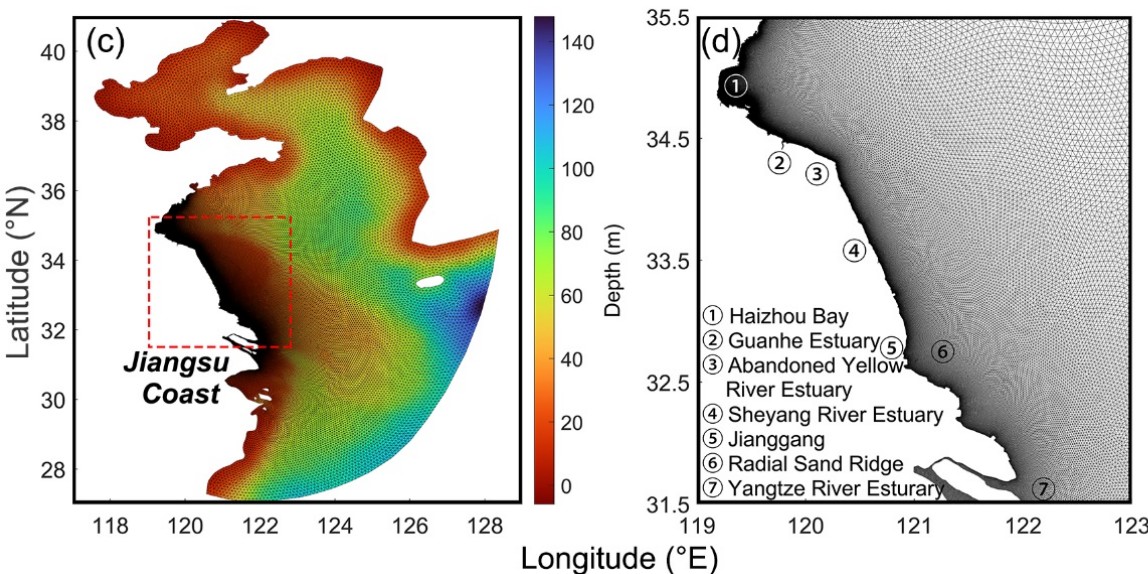

**Figure 1.** (**a**) Typical typhoon tracks and intensity diagram affecting the JC. The polygonal area enclosed by the red line in Figure 1a is Jiangsu Province, China. The area surrounded by a black dashed rectangle is the JC. (**b**) The bathymetry of the JC and the verification points within the region. (**c**) Grids and bathymetry of the model domain. The region represented by the red dashed box is the JC. (**d**) The grids within the JC region. The numbers represent the corresponding place names.

## 3. Materials and Methods

### 3.1. Data Collection

We collected validation data during the passage of No. 1909 Typhoon Lekima from July to August 2019 to calibrate the model parameters. The water level and current validation data were obtained from the National Marine Data Center (https://mds.nmdis.org.cn/ (accessed on 18 June 2024)). The wave validation data and storm surge–wave analysis dataset were obtained from Yang et al. [1] and Zhang et al. [21], respectively. Due to the lack of field sediment observation data, the model was calibrated using the Geostationary Ocean Color Imager (GOCI) data (https://kosc.kiost.ac.kr/ (accessed on 18 June 2024)). Moreover, hourly sea-surface wind forcing data were downloaded from the fifth-generation ECMWF atmospheric reanalysis of the global climate (ERA5) (https://cds.climate.copernicus.eu (accessed on 18 June 2024)) to capture the dynamic processes during typhoons' passage.

### 3.2. Numerical Model

The FVCOM [22] is a three-dimensional hydrodynamic model that uses an unstructured mesh and a σ-stretched coordinate system. The model was applied in this study to simulate hydrodynamics and sediment dynamics on the JC during TCs' passage. The model was configured as the modified Mellor and Yamada level 2.5 (MY-2.5) [23] for vertical mixing and the Smagorinsky [24] turbulent closure scheme for horizontal mixing. It is adept for the complex coastlines of coastal areas [25–28]. The SWAN model, by Booij et al. [29] was employed to simulate wind-wave generation and propagation processes. In addition, the sediment module used in this study was FVCOM-SED, which has been extensively applied in coastal areas [19,30–32]. The wave model SWAN was run separately, and the generated wave field was input into the FVCOM, which is called one-way wave–current coupling.

### 3.3. GOCI Satellite Remote Sensing Suspended Sediment Concentration (SSC) Retrieved Algorithm

The GOCI is a sensor mounted on the world's first geostationary orbit ocean color satellite, the Communication, Ocean, and Meteorological Satellite (COMS), launched by South Korea. The central longitude and latitude of the GOCI image is 130° E and 36° N, respectively. The coverage area is 2500 km × 2500 km, and the spatial resolution is 500 m.

The GOCI has a time resolution of 8 scenes per day from 0 to 7 h UTC. Due to its high temporal resolution, GOCI data have been widely used in the studies of SSC distribution and transport in nearshore areas [33].

We selected cloud-free images of the JC area, captured on 20 July and 2 August 2019, to verify the modeled surface SSC. This study used the retrieved algorithm for sea-surface SSC by He et al. [34], which is widely used in high-SSC areas [27,33]. The retrieved surface SSC algorithm is as follows:

$$SSC = 10^{1.0758 + 1.1230 \times \frac{ref(745 \text{ nm})}{ref(490 \text{ nm})}} \tag{1}$$

where *SSC* indicates the suspended sediment concentration (unit: mg/L), while $ref(745 \text{ nm})$ and $ref(490 \text{ nm})$ are the remote sensing reflectance of the 745 and 490 nm bands of the GOCI satellite, respectively.

### 3.4. Model Configuration

3.4.1. Model Setup and Initial Conditions

The model domain is depicted in Figure 1. We employed the Surface-water Modeling System (SMS) to create grids based on the 2022 Google Earth coastline data. The bathymetry data in the JC area were obtained from the 2019 electronic chart, while bathymetry data for the remaining area in the model domain were obtained from the ETOPO1 Global Relief Model, which is at 1 arc-minute spatial resolution. The grid resolution within the JC ranges from 100 m to 300 m nearshore and was, 10 km along the opening boundary. The model employed a uniformed sigma-stretched coordinate system consisting of six vertical layers. In order to drive the hydrodynamic model along the open boundary, the NAO.99b short-period ocean tide model was utilized, considering 16 major constituents (M2, S2, K1, O1, N2, P1, K2, Q1, M1, J1, OO1, 2N2, Mu2, Nu2, L2, and T2) [35]. Furthermore, identical grids and winds were applied in the SWAN model to simulate the wave propagation within the model domain. The wave spectrum was calculated in the 0.031–0.548 Hz range and divided into 40 frequency steps according to the logarithmic distribution. The rest of the parameters were given the default settings based on the SWAN manual.

According to the General Report on Comprehensive Survey and Evaluation of Offshore Oceans in Jiangsu Province [15], there are significant differences in the zoning of sediment in the JC area. The median particle size distribution of sediment in Haizhou Bay, the abandoned Yellow River Estuary, the radial sand ridge, and the Yangtze River Estuary ranges from 0.01 to 0.08 mm. Therefore, the setting of sediment parameters in the model has spatial heterogeneity. Based on the relationship between suspended sand particle size and sedimentation velocity measured experimentally, the settling velocity was set to 0.3~3.8 mm/s [36]. According to the empirical formula of particle size and erosion rate, the erosion rate was set to $2 \times 10^{-5}$~$5 \times 10^{-5}$ kg/m$^2$/s. The critical erosion stress was set to be 0.6~0.12 N/m$^2$, while the critical deposition stress was set at a constant value of 0.05 N/m$^2$ (Table 1).

**Table 1.** Main parameters used in the coupled model.

| Model Parameters | Value |
|---|---|
| Number of mesh nodes and elements | 92,068, 178,933 |
| Model external time step | 1.0 s |
| Model internal time step | 10 s |
| Median grain size | 0.01–0.08 mm |
| Settling velocity | 0.3–3.8 mm/s |
| Critical erosion stress | 0.6–0.12 N/m$^2$ |
| Critical deposition stress | 0.05 N/m$^2$ |
| Erosion rate | $2 \times 10^{-5}$–$5 \times 10^{-5}$ kg/m$^2$/s |
| Porosity | 0.5 |

### 3.4.2. Numerical Experiment Settings

A total of 11 numerical experiments were designed to explore different factors, such as typhoon tracks, intensities and tidal superposition conditions, in the impact of hydrodynamics and sedimentary processes on the JC. In the first four experiments, we selected four typhoons between 2000 and 2020 that are representative of the four types of typhoons discussed earlier. The track of No. 1909 Typhoon Lekima represents a typical example of passing through the JC after making landfall in the south of the JC; the track of No. 1210 Typhoon Damrey is representative of landfall along the northern coast of the JC; No. 1810 Typhoon Ampil's track is a typical path of landfall along the southern JC; No. 2008 Typhoon Bavi passed through the Yellow Sea and affected the JC, making it a typical representative of this type of typhoon. Among them, experiments 1–4 were designed to explore the influence of four typical typhoons, with tides and waves taken into account. Experiments 5–8 showed the situations when the wind forcing of four typical typhoons is amplified by 1.3 times. Experiment 9 simulated the conditions of No. 9711 Typhoon Winnie, which had a significant impact on the JC. Experiment 10 excluded the tidal effects and only considered the effects of wind forcing and waves from Lekima on the hydrodynamics and sediment dynamics. Experiment 11 assumed that No. 1909 Typhoon Lekima occurred during the astronomical spring tide period. All experiments were preheated for 10 days before starting, without considering the effects of runoff, temperature, and salinity. The configuration description of the experiment is summarized in Table 2.

**Table 2.** Experiments' descriptions.

| Experiment | Description |
|---|---|
| Exp.1 | Forced by wind, tidal currents, and waves during No. 1909 Typhoon Lekima |
| Exp.2 | Same as Exp.1, but during No. 2008 Typhoon Bavi |
| Exp.3 | Same as Exp.1, but during No. 1810 Typhoon Ampil |
| Exp.4 | Same as Exp.1, but during No. 1210 Typhoon Damrey |
| Exp.5 | Same as Exp.1, but the wind force was amplified 1.3 times |
| Exp.6 | Same as Exp.2, but the wind force was amplified 1.3 times |
| Exp.7 | Same as Exp.3, but the wind force was amplified 1.3 times |
| Exp.8 | Same as Exp.4, but the wind force was amplified 1.3 times |
| Exp.9 | Same as Exp.1, but during No. 9711 Typhoon Winnie |
| Exp.10 | Same as Exp.1, but without tidal effects |
| Exp.11 | Same as Exp.1, but during the spring tidal cycle |

### 3.4.3. Model Validation

As shown in Figure 2a–h, the observed and modeled tidal elevation, tidal currents, and waves were compared to calibrate the model. The water elevation data were collected from 25 July to 24 August 2019 at LS (Lvsi), SYHK (Sheyang Hekou), and LYG (Lianyungang). The tidal current data were measured at stations DSYC and LYGHX (Figure 1b) during the neap tide from 00:00 on 26 to 27 July 2019, and during the spring tide from 00:00 on 3 to 4 August 2019. The wave data were collected from 22:00 on 6 August to 2:00 on 13 August 2019 at station W. In addition, we obtained a wave reanalysis dataset during storm events in the Yellow Sea for wave height verification.

The sediment module was calibrated using remote sensing (GOCI) data. Eight cloud-free images captured from 00:00 to 07:00 (UTC) on 2 August 2019 were selected to verify the magnitudes and trends of the modeled surface SSC at the 8 stations (S1–S8), as shown in Figure 2i–p. After the model was calibrated, we further verified it with the retrieved surface SSC fields on different days and times, at 04:00 on 20 July (UTC) and 04:00 on 2 August (UTC) (Figure 3a–d).

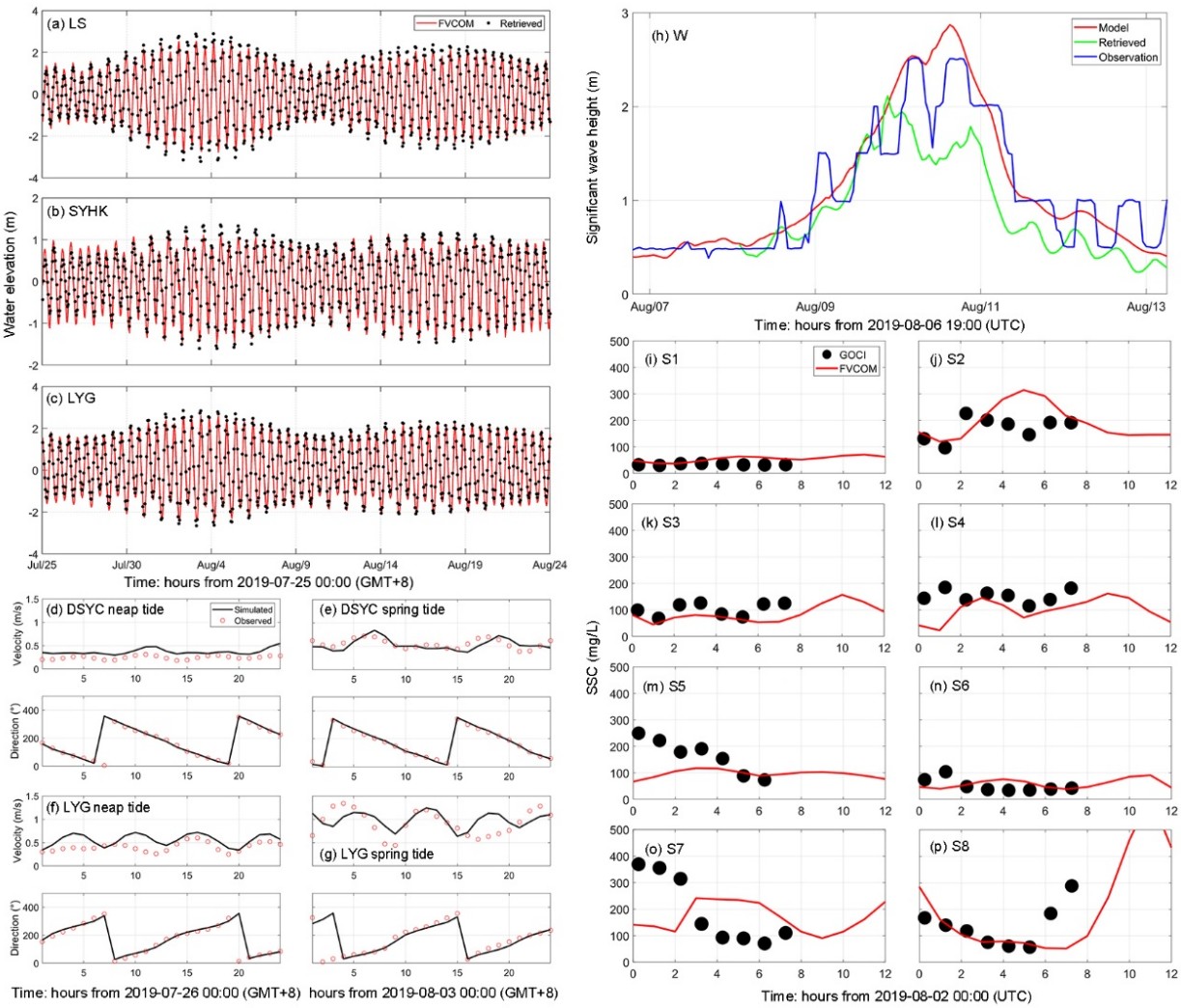

**Figure 2.** From 25 July to 24 August 2019, hourly water-level validation at three tidal stations: (**a**) LS, (**b**) SYHK, and (**c**) LYG. (**d**–**g**) verification of currents at DSYC and LYG during spring and neap tides, respectively. (**h**) At site W, the validation between the simulated significant wave height (red line) of SWAN, the obtained wave reanalysis dataset (green line), and the measured values (blue line). (**i**–**p**) Hourly comparison between the GOCI-retrieved and the simulated SSC after 0:00 (UTC) on 2 August 2019, at the S1–S8 SSC verification point.

To verify the accuracy of the coupled model for hydrodynamics and sediment dynamics, the root-mean-square error (RMSE) and correlation coefficient (CC) [37] for tidal elevation, tidal currents, waves, and SSC were calculated as shown in Table 3. The formulae for the statistics are as follows:

$$RMSE = \sqrt{\frac{1}{n}\sum_{i=1}^{n}(m_i - o_i)^2} \tag{2}$$

$$CC = \frac{1}{n}\sum_{i=1}^{n}\frac{(m_i - \overline{m})(o_i - \overline{o})}{S_m S_o} \tag{3}$$

where $n$ is the number of observations and $m_i$ and $o_i$ represent the modeled and observed values at time $t$, respectively, which have the mean values $\overline{m}$ and $\overline{o}$, and the standard deviations $S_m$ and $S_o$, respectively. The smaller the *RMSE*, the smaller the deviation between the modeled and observed values. A value of *CC* close to 1 suggests a strong agreement between the observed and modeled values.

**Table 3.** Error statistics of the model results.

| Variable | Station | CC | RMSE |
|---|---|---|---|
| Water level | LS | 0.97 | 0.37 m |
| | SYHK | 0.95 | 0.21 m |
| | LYG | 0.97 | 0.33 m |
| Current velocity | DSYC neap tide | 0.60 | 0.13 m/s |
| | DSYP spring tide | 0.55 | 0.10 m/s |
| | LYG neap tide | 0.54 | 0.18 m/s |
| | LYG spring tide | 0.39 | 0.26 m/s |
| Significant wave height | W | 0.89 | 0.33 m |
| Sea-surface SSC | S1 | −0.23 | 19.85 mg/L |
| | S2 | 0.18 | 85.13 mg/L |
| | S3 | 0.17 | 43.34 mg/L |
| | S4 | −0.02 | 77.37 mg/L |
| | S5 | −0.21 | 162.10 mg/L |
| | S6 | −0.52 | 32.98 mg/L |
| | S7 | −0.75 | 165.31 mg/L |
| | S8 | 0.03 | 104.84 mg/L |

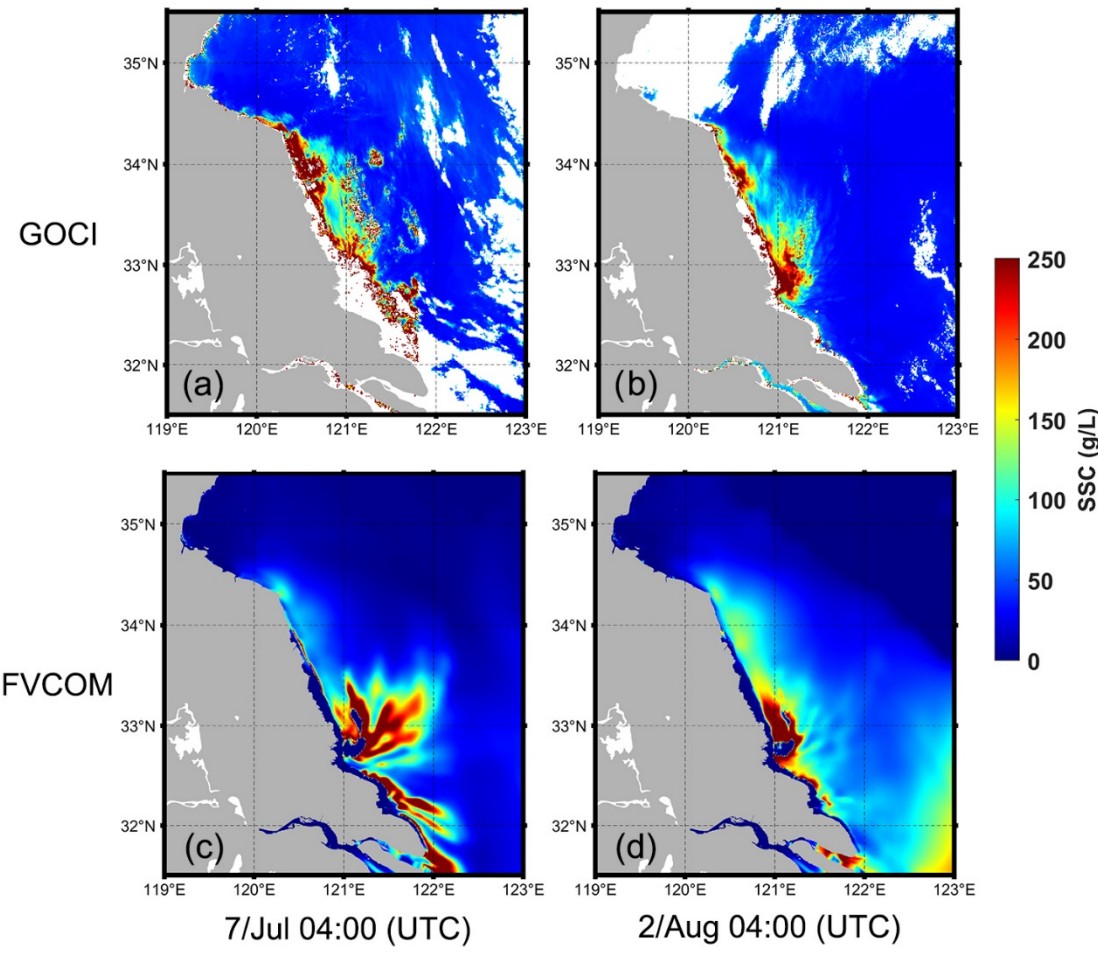

**Figure 3.** Distribution of sea-surface SSC on the JC obtained from (**a**,**b**) GOCI-retrieved and (**c**,**d**) FVCOM-modeled data at 04:00 (UTC) on 20 July and 04:00 (UTC) on 2 August 2019, respectively.

The statistics of the modeled results (Table 3) showed a strong correlation with the observed water level and significant wave height values, with CCs higher than 0.89. The CC of current magnitude and direction could be categorized as moderate (0.4–0.6), with the

RMSE being lower than 0.26 m/s. The CC value of SSC near the surface level was relatively low, but the RMSE was within an acceptable range (20–165 mg/L). The modeled surface SSC distribution pattern was consistent with the GOCI data. In addition, to strengthen the robustness of the model performance, we incorporated historical TC field data into model validation (Appendix A). The results indicated that the coupled model demonstrated satisfying performance in simulating the regional ocean dynamics and sediment transport.

## 4. Results

### 4.1. The Response of Hydrodynamics and Sediment Dynamics to Typhoon Lekima

Figure 4 shows the differences in hydrodynamics and sediment dynamics under fair-weather and Typhoon Lekima conditions. The right-column in the graph represents the time when Typhoon Lekima significantly impacted the JC (10 August 2019 23:00 UTC). To eliminate the deviation of tidal effects on currents and SSC, the similar tidal period that occurred 14 days prior when the wind speed was relatively low (26 July 2019 22:00 UTC) was selected as the fair-weather conditions.

Under the fair-weather conditions, the sea-surface wind speed was relatively low, generally below 5 m/s, with the significant wave height (Hs) around 0.5 m. At the moment when the typhoon made a significant impact, the center of the TC was located on the land of Jiangsu Province, with the wind direction rotating counterclockwise. The lower friction force on the sea surface allowed for higher southeasterly wind speeds of over 20 m/s. Correspondingly, the Hs also significantly increased compared to fair-weather conditions, reaching over 5 m in the outer sea area. The Hs gradually decreased towards the nearshore, with smaller waves in the RSRs, but could still reach a height of about 1 m (Figure 4(a1,a2)).

At this moment, under fair-weather conditions, the direction of the surface and bottom currents was essentially the same, with the tidal currents converging centripetally in the RSRs. Here, the SSC was slightly higher but generally below 0.25 g/L. However, the flow field was very different during the typhoon compared to fair-weather conditions at a similar tidal period. The strong southerly wind caused by the typhoon enhanced the southeast direction currents on the south side of the RSRs, with a current velocity of over 1 m/s. In areas with higher wind speeds, the effect on the surface currents was more significant. Correspondingly, there was a significant increase in the surface SSC compared to the fair-weather conditions, and there was a high-value area of SSC in the entire Jiangsu coastal area, which could reach 1.5–2 g/L on the southern side. On the north side of the RSRs, the bottom layer's flow velocity differed significantly from the fair-weather conditions, shifting from northwest to southeast. The SSC in the bottom layer was higher than that in the surface layer, generally between 1 and 2 g/L, and could reach over 3 g/L in the area from the abandoned Yellow River Estuary to the Sheyang River Estuary, where the wave height decreased rapidly. The strong wave energy dissipation in shallow water greatly increased the shear stress on the seabed, enhanced the process of sediment resuspension and, thus, significantly increased the SSC (Figure 4(b1,b2,c1,c2)).

Figure 5 shows the results of the wind, current, significant wave height, bottom stress, water level, SSC, and their vertical time-series distribution located at point W near the RSR area during a tidal cycle before and after the passage of Typhoon Lekima. During the typhoon period, the wind speed significantly increased, reaching about 20 m/s in the southeast direction. The prevailing wind direction before the typhoon passed was mainly southerly, and it shifted to northerly after the passage. During the period of the typhoon's impact, the sea currents, originally dominated by tidal effects, underwent significant changes, with a noticeable increase in the current velocity. At the peak flow velocity, the currents flowed from southwest to northeast. During the passage of the typhoon, the significant wave height reached 4 m, whereas, under fair weather conditions, it generally did not exceed 1 m. Correspondingly, both the bottom stress and SSC exhibited noticeable increases during the typhoon's passage. Originally, the SSC and bed shear stress were significantly correlated with the tidal cycles. However, Typhoon Lekima occurred during a period of neap tides, during which the relatively small bottom stress increased significantly.

The sustained high bottom stress resulted in a significant increase in SSC, with the SSC in the bottom layer being greater than that in the surface layer. The depth-averaged SSC could reach 1 g/L during the typhoon's passage, nearly a hundredfold increase compared to SSC levels during neap tides under fair-weather conditions. The peak SSC occurred during the flood tide, which was related to the combined effect of a large wave current induced bottom stress at that moment. Following the passage of the typhoon, both SSC and bottom shear stress rapidly returned to normal levels. Furthermore, during the impact of Lekima, there was a sequence of stormwater levels decreasing, followed by an increase due to the impact of Lekima. The water level increase slightly exceeded the decrease, reaching around 0.5 m. In the situations of low wind speeds, the influence of wind forcing on water levels was almost negligible.

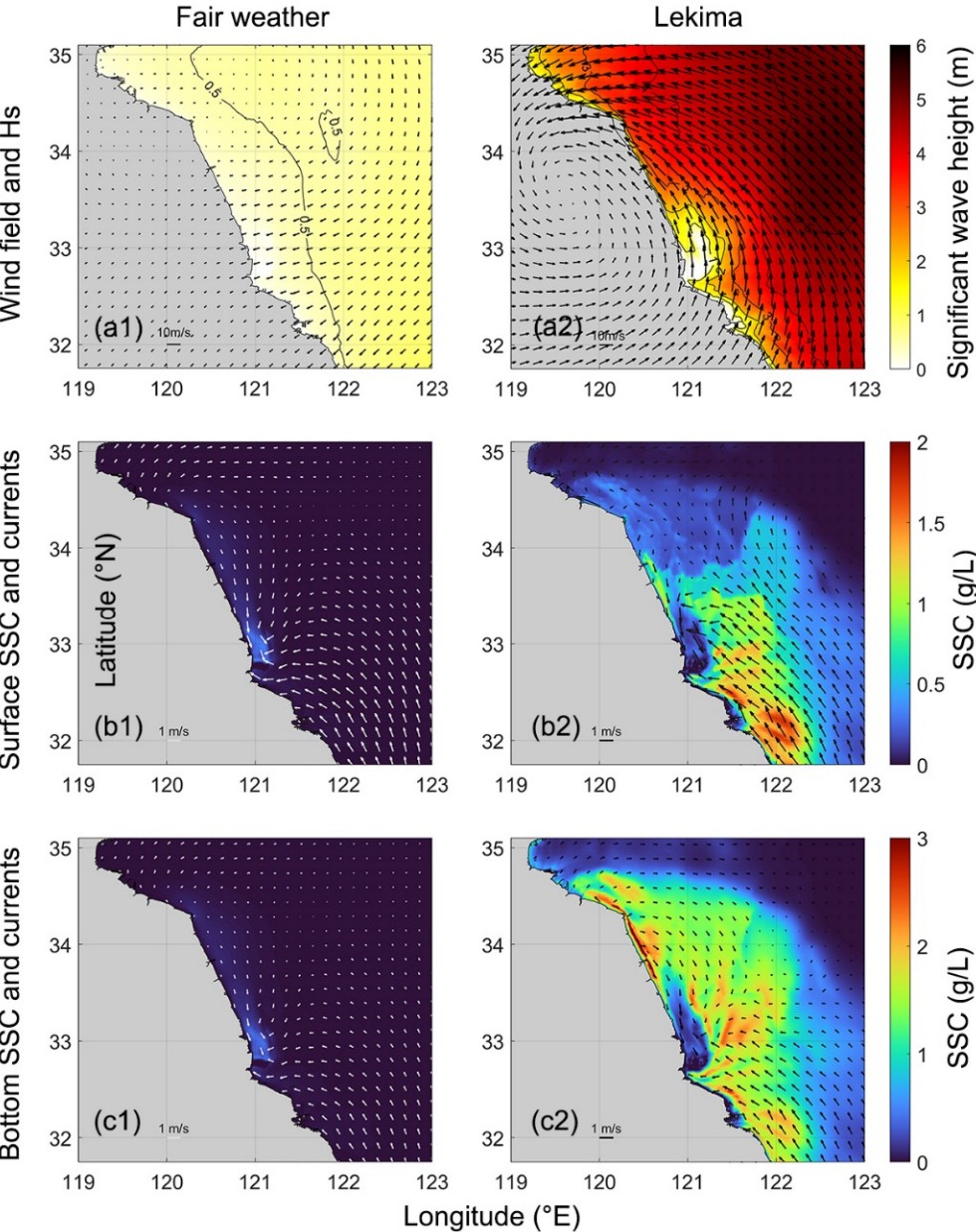

**Figure 4.** (**a1**,**a2**) The significant wave height and wind field, (**b1**,**b2**) surface currents with SSC, and (**c1**,**c2**) bottom currents with SSC at a similar tidal period during the passage of Typhoon Lekima (23:00 10 August) and under fair-weather conditions (22:00 26 July).

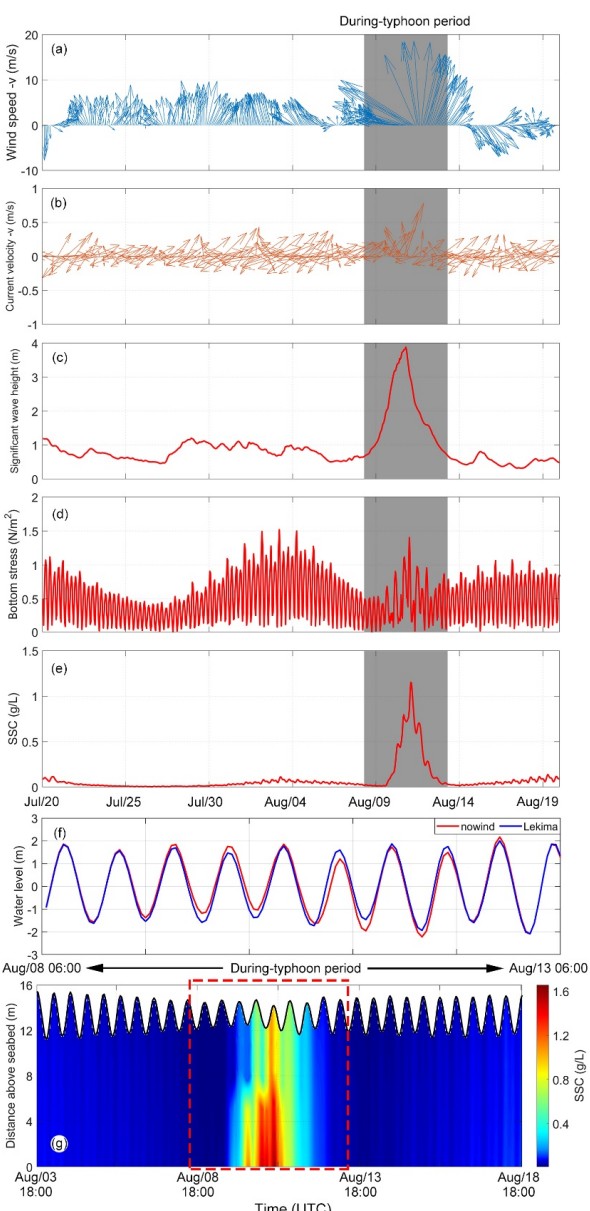

**Figure 5.** During a tidal cycle before and after the passage of Typhoon Lekima: (**a**) wind speed in the direction of v, (**b**) current velocity in the direction of v, (**c**) significant wave height, (**d**) bottom stress, (**e**) SSC, (**f**) comparisons of water levels under no-wind conditions and Typhoon Lekima, and (**g**) SSC vertical profile time series. The gray area and the red dashed box represent the period of typhoon transit impact.

### 4.2. The Impact of TCs with Different Tracks and Intensities on Sediment Dynamics

To explore the impact of TCs with different tracks and intensities on the sediment dynamics on the JC, we selected four representative tracks of typhoons that affected the JC. During the passage of four typical typhoons, except for Typhoon Bavi, the sediment flux induced by the other three typhoons on the JC was directed northwestward from the southeast along the coastline. This pattern was attributed to the prevailing easterly or southeasterly wind anomalies during the passage of these typhoons. However, as Bavi moved northward out of the Yellow Sea, with the JC nearshore on its left, the prevailing wind anomalies were mainly northwesterly, and the magnitude of the wind anomalies was relatively small. Consequently, the sediment flux induced by Bavi was relatively minor, reaching a maximum of only 0.2 kg/m/s (Figures 6d and 7(d1)).

The magnitudes and patterns of sediment flux induced by Typhoon Lekima and Typhoon Damrey were quite similar. The area near LS and the RSRs exhibited high sediment flux values, reaching over 0.8 kg/m/s. However, compared to Damrey, the wind anomalies associated with Lekima were larger in the offshore areas with a maximum of 7 m/s, and relatively smaller nearshore wind of below 3 m/s in the prevailing southeasterly direction. In contrast, the wind anomalies associated with Damrey ranged between 4 and 6 m/s, with easterly winds in the southern part of the nearshore area and southeasterly winds in the northern part. The wind anomaly pattern of Damrey was similar to that of Ampil, but the magnitude was slightly smaller between 3 and 5 m/s. Correspondingly, the sediment flux induced by Ampil was of a smaller magnitude compared to that induced by Damrey and Lekima. Additionally, during the passage of Ampil, the high sediment flux values were distributed in the offshore areas of the RSR, with no high-value area near LS, and the overall magnitude of the sediment flux ranged between 0.1 and 0.5 kg/m/s (Figures 6a–c and 7(a1–c1)).

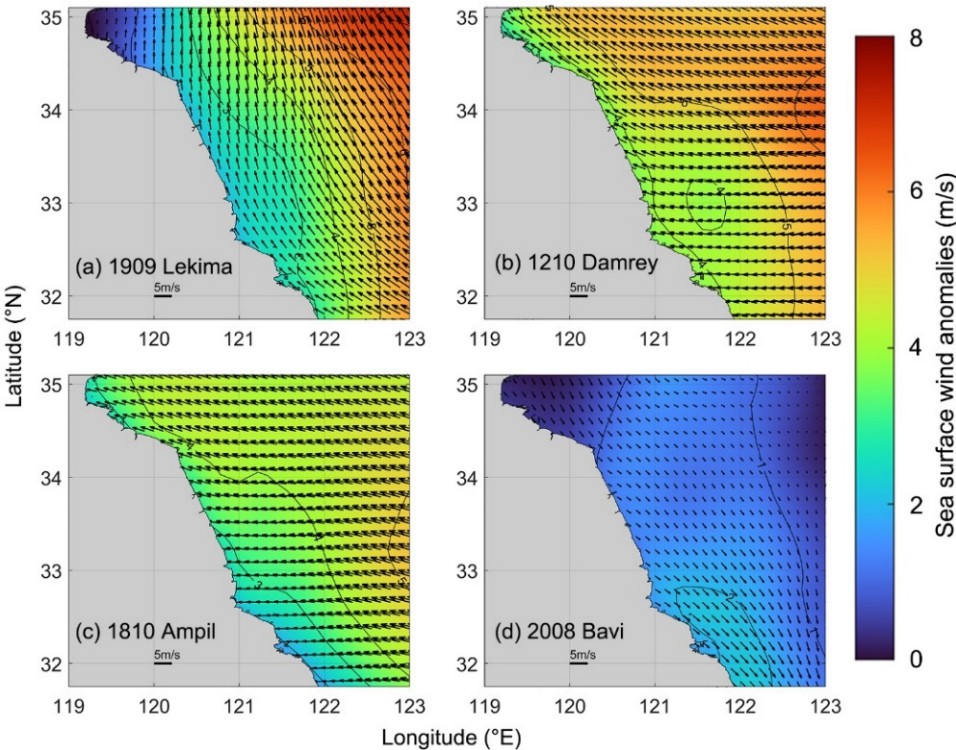

**Figure 6.** Typhoons (**a**) Lekima, (**b**) Damrey, (**c**) Ampil, and (**d**) Bavi induced wind anomalies in 5 days from the 30-year average (1991–2020) wind field on the JC. The arrow represents the direction of the wind vector that deviated from the average wind field during the typhoon, and the color chart illustrates the wind anomaly.

After the passage of these four typhoons, the model results indicated certain changes in the sediment thickness of the seabed on the JC. The most prominent feature was that the seabed exhibited a striped pattern with strong erosion/deposition closely related to the seafloor topography. Particularly in the RSRs, seabed erosion often occurred over the sand ridges where the water depth is relatively shallow, while sediment accretion occurred in deep channels. This was due to the barrier effect of the sand ridges, where suspended sediment particles in the ocean water column in the RSRs were captured and accumulated in the deep channels, whereas, within the sand ridges, the strong shear stress from the combined action of waves and tidal currents enhanced the sediment resuspension process, leading to seabed erosion. The morphological changes induced by Bavi were minimal, with variations of less than 0.1 m, which was closely related to the magnitude of sediment flux. On the other hand, the bed thickness changes caused by Damrey were the most

significant, with erosion/deposition in sand ridges and deep channels exceeding ±0.3 m. The distribution pattern of seabed erosion/siltation caused by Ampil and Lekima was similar, but that of Ampil was slightly larger, with a maximum of around ±0.2 m, while that of Lekima was within ±0.15 m (Figure 7(a2,d2)).

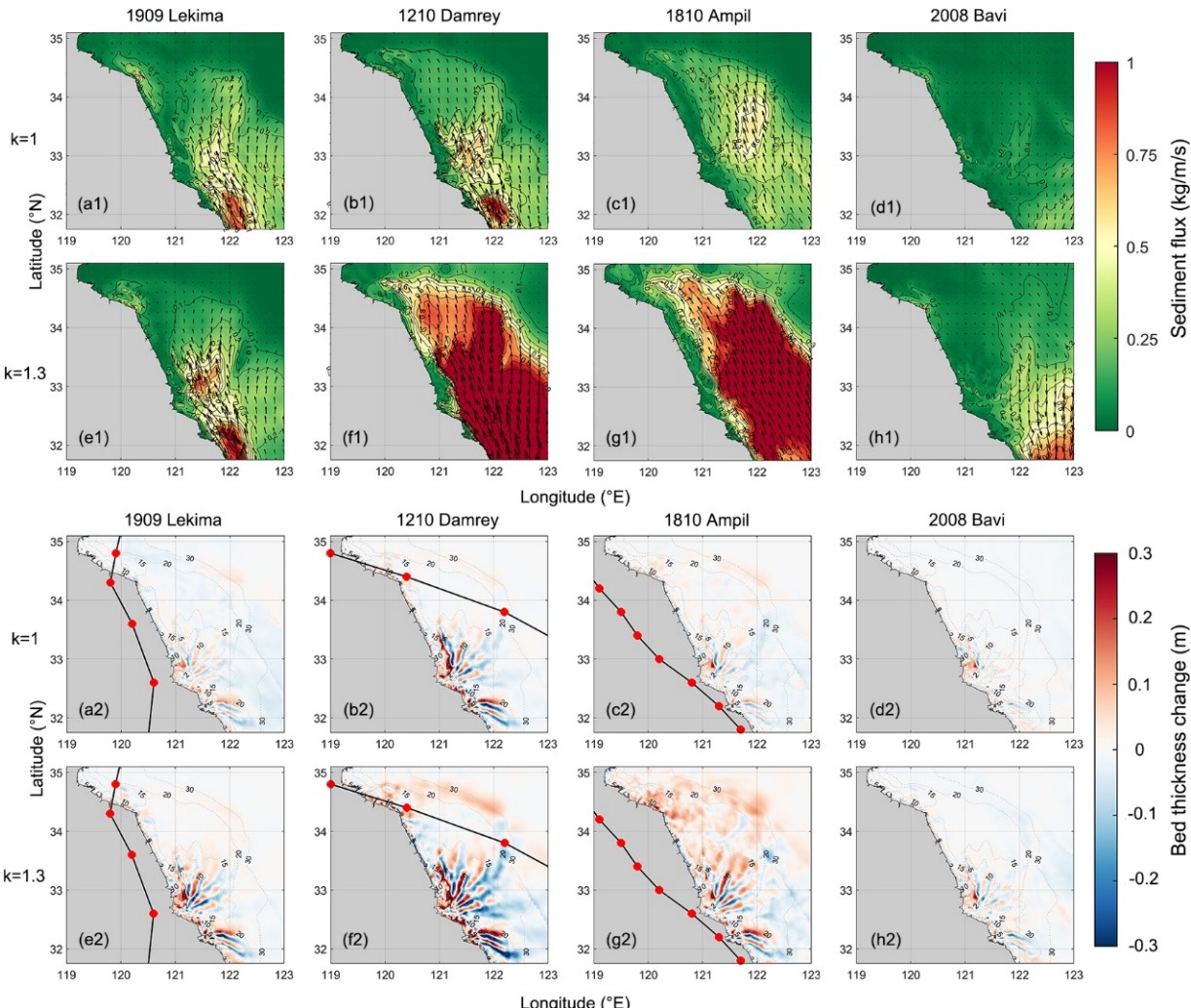

**Figure 7.** (**a1–d1**) The sediment flux in the JC area during a tidal cycle under the influence of Typhoons Lekima, Damrey, Ampil, and Bavi. The first-row k = 1 represents the actual typhoon wind field, and the wind field in the second row (**e1–h1**) with k = 1.3 corresponds to the wind field of the typhoon when increased by 1.3 times. (**a2–h2**) The bed thickness changes under the influence of the corresponding typhoon and enhanced wind field, where the red dots and black lines represent the track of the corresponding typhoon, and the dashed lines represent the water depth.

After the typhoon wind field was amplified by 1.3 times, the sediment flux in the JC area significantly increased, especially with Damrey and Ampil, and large areas of high sediment flux exceeding 1 kg/m/s appeared throughout the nearshore waters. The direction of sediment flux remained largely unchanged. Compared to the original conditions, the increase in sediment flux caused by the strengthened wind field of Lekima was not significant, with only high-value areas of around 1 kg/m/s appearing near LS. The impact of the strengthened wind field of Bavi on the sediment flux near the north side of the Yangtze River Estuary was significant, with the sediment flux towards the south exceeding 0.75 kg/m/s. Under the influence of the strengthened wind fields of Damrey and Ampil, sediment accretion occurred in the seabed of the Yellow Sea north of 33° N, while significant seabed erosion occurred to the south of 33° N. The distribution pattern of

erosion and deposition in the RSRs remained largely unchanged, but there was a noticeable increase in seabed thickness changes, especially with Damrey, during which there were large areas of ±0.3 m variation near sand ridges and deep channels. Ampil was slightly weaker but still had a significant impact on the seabed morphology. The strengthening of the wind fields by Lekima and Bavi also increased the change in seabed thickness by about 50% (Figure 7(e1–h1,e2–h2)).

The above results indicate that typhoons landing in or passing through Jiangsu often bring sediment flux from southeast to northwest along the JC, and that the sediment flux increases with the typhoon wind speed. Conversely, typhoons passing northward in the Yellow Sea's offshore area can cause sediment flux from north to south, with a weaker impact on sediment dynamics if the typhoon's center is relatively far from land. In terms of changes in bed morphology, typhoons induce strong stripe erosion/deposition at sand ridges and deep channels, with the intensity of erosion and deposition closely related to the typhoon's intensity.

## 5. Discussion

### 5.1. Example of a TC Affecting the JC during the Astronomical Tidal Period

No. 9711 Typhoon Winnie, with strong winds and heavy rainfall, landed on the eastern coast of China on 18 August 1997. Coupled with astronomically high tides, the typhoon induced a severe storm surge in the offshore areas of the JC. Both Nantong Port and Lianyungang Port experienced record-breaking water levels. Significant coastal erosion and sedimentation and extensive seawall damage, resulted in substantial economic losses [8,38].

The results shown in Figure 8I indicate that the sediment flux caused by Typhoon Winnie was significantly higher than that by the four typical typhoons discussed earlier. There existed a high-value area of sediment flux in the RSRs, reaching up to over 1.6 kg/m/s, in the direction along the coastline from southeast to northwest (Figure 8I(a)). The distribution of sediment flux was closely related to the distribution of wind anomalies. The wind anomalies were also southeastward and the magnitude decreased from southeast to northwest in the range between 2 m/s and 5 m/s (Figure 8I(b)). However, compared with other typhoons, the difference in wind anomalies was not significant, indicating that the large sediment flux was not solely caused by the intensity of the typhoon.

Regarding the morphological changes in the seabed, strong striped erosion and deposition patterns were exhibited at sand ridges and deep channels, with a maximum magnitude of approximately ±0.3 m (Figure 8I(c)). This finding closely aligns with the results of Yazhen et al. [39] that an average sediment deposition was about 20 cm in the southern port and northern passage of the Yangtze River Estuary during Typhoon Winnie. Furthermore, the geomorphological changes induced by Winnie were significantly greater than those caused by the four typical typhoons, approaching the results of Typhoon Lekima with a 1.3-fold wind increase, and only slightly weaker than the results after Typhoon Damrey with a 1.3-fold increase in wind field strength.

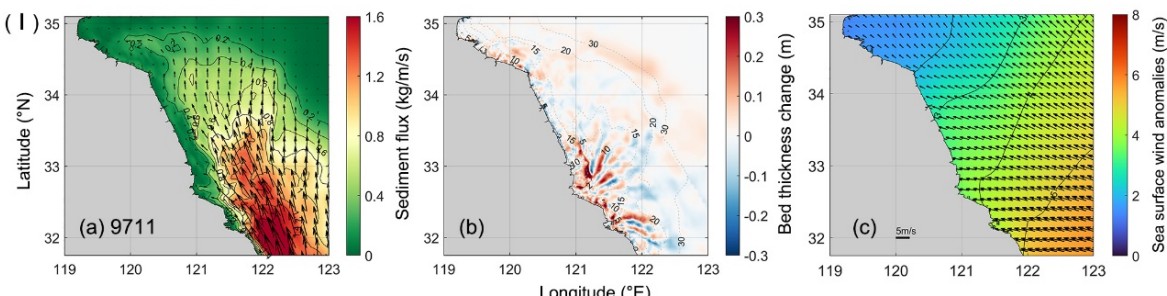

**Figure 8.** *Cont.*

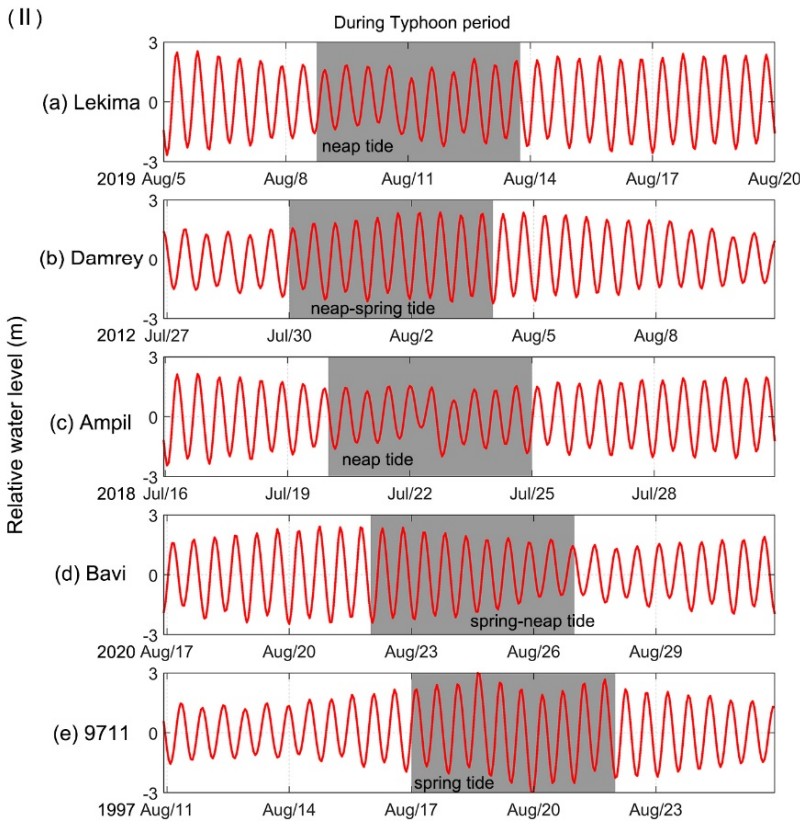

**Figure 8.** (**I**) During the tidal cycle of Typhoon Winnie's impact period, (**a**) sediment flux, (**b**) bed thickness changes, and (**c**) sea-surface wind anomalies over the 5 days affected by the typhoon. (**II**) Time series of relative water levels during typical typhoons (**a**) Lekima, (**b**) Damrey, (**c**) Ampil, (**d**) Bavi, and (**e**) Winnie affecting the JC at point W.

### 5.2. Tidal Conditions during Different Typhoon Impact Periods

Figure 8II illustrates the tidal conditions during different typhoon impact periods. The passage of Typhoons Lekima and Ampil was in the neap tide cycle, and the diurnal tidal range was relatively small. However, some fluctuations in tidal periodicity occurred, manifesting as storm surges and recessions. Typhoons Bavi and Damrey corresponded to periods between neap tide and spring tide and between spring tide and neap tide, respectively, with tidal periodicity showing more regular and uniform changes. The impact period of Typhoon Winnie coincided with astronomical high tides, resulting in extreme storm surges and recessions. The maximum tidal ranges could exceed 5 m (Figure 8II(a–e)). Larger tidal ranges led to increased tidal current velocity, thereby increasing the bed shear stress which promoted sediment resuspension processes, leading to a significant increase in SSC. The increased current velocity and SSC further intensified the sediment transport induced by the typhoon. This was a significant factor in why the sedimentary dynamic process was affected by Typhoon Winnie more severely compared to the other typhoons.

### 5.3. The Impact of Different Tidal Conditions on Sediment Dynamics during Typhoons

Figure 9 indicates that if Typhoon Lekima occurred during spring tides, the sediment flux towards the northwest direction along the coast would significantly increase. In the RSRs, there was a high-value zone of sediment flux exceeding 1 kg/m/s, approximately 80% to 100% higher than that during neap tides. If it was solely driven by wind forcing without tidal effects, the sediment flux was minimal, with a maximum of about 0.3 kg/m/s, predominantly in the northerly direction, which was significantly lower compared to neap tide conditions (Figure 9(a1–c1)). Correspondingly, nearshore bed thickness changes were closely associated with the magnitude of sediment flux. If there was no tidal forcing, the

variations in bed thickness were minimal, with a maximum change of around ±0.15 m. In Exp.1 (during the neap tide), the extent of geomorphological changes was larger than that in Exp.10 (without tidal influence), although the bed thickness changes were relatively smaller. However, in Exp.11 (during the spring tide), the bed thickness changes significantly increased, with the maximum change reaching up to ±0.3 m, closely resembling the sediment flux and geomorphological changes induced by Typhoon Winnie (Figure 9(a2–c2)).

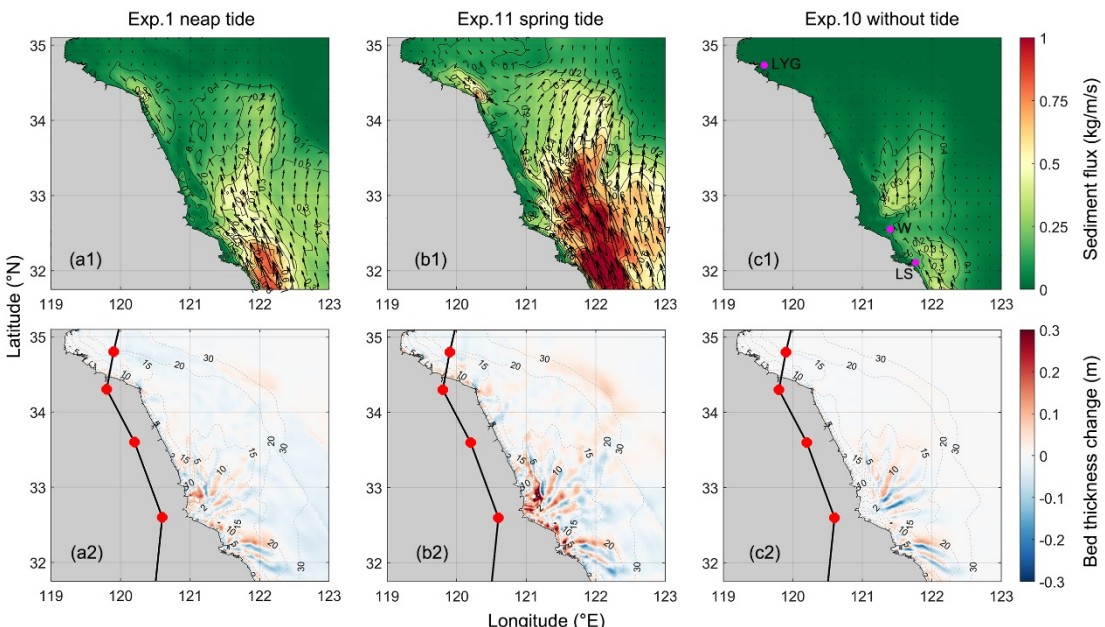

**Figure 9.** (**a1**,**a2**) During the neap tide period, (**b1**,**b2**) during the spring tide period, and (**c1**,**c2**) without tidal open-boundary conditions, the sediment flux (row 1) and bed thickness changes (row 2) during the tidal cycle of Typhoon Lekima affecting the JC.

Figure 10 illustrates that, at points LS, LYG, and W, during both the spring and neap tides, SSC and bottom shear stress exhibited regular cyclic variations under tidal effects. At the moments of peak flood and peak ebb, both the bottom stress and SSC were relatively high, while they were lower during the high-water and low-water moments. Throughout the spring tide period, the SSC and bottom stress at these points were notably higher compared to the neap tide period. A significant increase was seen at these points during the typhoon's passage. Specifically, LS and LYG exhibited maximum bottom-layer SSC between 1.4 and 1.7 g/L, and the maximum bottom stress between 1.5 and 2 N/m$^2$ during spring tide, while the SSC and bottom stress at point W, located in the RSR area, reached up to 4.5 g/L and 3 N/m$^2$, respectively. During the neap tide, LS and W experienced maximum SSC at around 1 g/L and 2 g/L, respectively, approximately 40–60% of the levels during spring tide, while the bottom stress was about 50–70% of that during spring tide. At LYG, the incremental changes in the SSC and bottom stress between spring tide and neap tide were not significant, with maximum SSC at around 1.3 g/L and maximum bottom stress at approximately 1.2 N/m$^2$. Conversely, in the absence of tidal effects, the bottom shear stress was generally below 0.5 N/m$^2$, and after the typhoon's passage the bottom stress was negligible. Even during the typhoon's passage, the SSC levels were below 0.1 g/L, which was less than 10% of those during neap tide (Figure 10a–f).

In summary, tidal effects play a fundamental and crucial role in the net sediment flux and geomorphological changes during typhoons' passage. Under the combined influence of the typhoon and tidal effects, the increase in bottom shear stress leads to elevated SSC, resulting in a larger sediment flux. Particularly during the spring tides, the net sediment flux induced by the typhoon can increase by 80–100% compared to the neap tides, accompanied by a significant rise in the bed thickness changes. These findings are consistent with the

conclusions of Fan et al. [40], that coastal erosion is more severe during the spring tides caused by a weak storm compared to the neap tides caused by a strong storm.

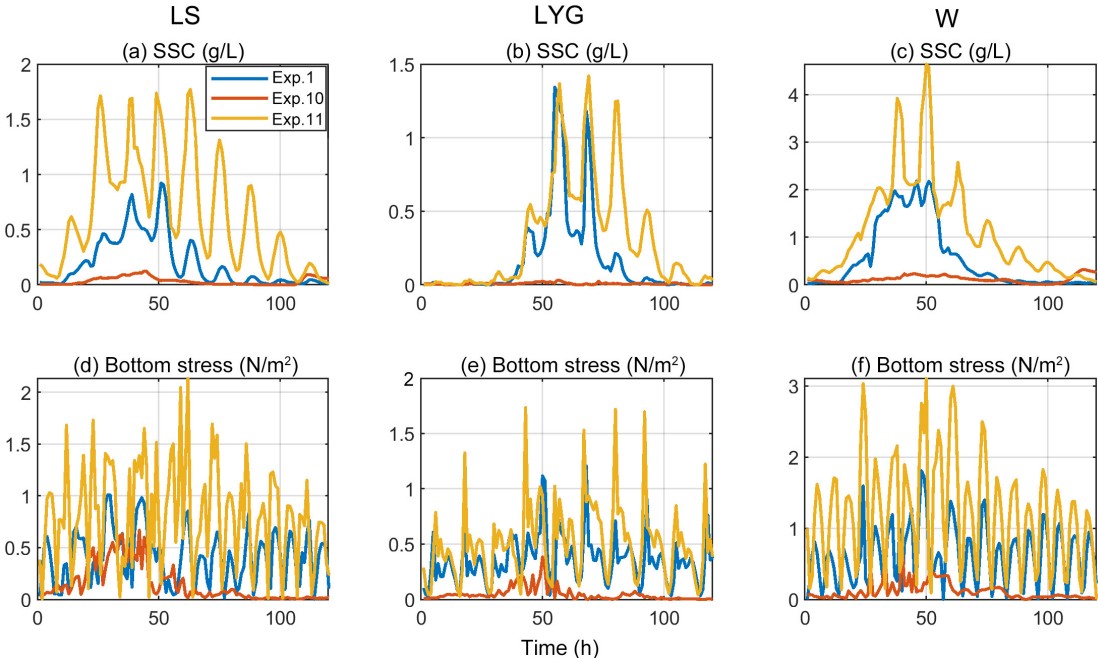

**Figure 10.** During the impact period of Typhoon Lekima, the time series of (**a**–**c**) bottom SSC and (**d**–**f**) bottom stress at points LS, LYG, and W under the conditions of Exp.1 (neap tide period), Exp.10 (without tide), and Exp.11 (spring tide period).

## 6. Conclusions

Based on the FVCOM hydrodynamic model, SWAN wave model, FVCOM-SED sediment module, this study constructed a 3D wave–current–sediment coupled numerical model for the Jiangsu Coast (JC) in China to investigate the response of sediment dynamics to TCs under different scenarios. The main conclusions of this study are as follows:

(1) Typhoons significantly influence the hydrodynamics and sediment dynamics of the JC. During the passage of Typhoon Lekima (2019), under the influence of southeasterly winds exceeding 20 m/s, the current velocity, bottom stress, and significant wave height all increased significantly, with the SSC reaching nearly one hundred times that under fair-weather conditions.

(2) Comparative analysis of four different typical-track typhoons showed that, under the influence of typhoons following three typical tracks and landing on the eastern coast of China, prevailing southeasterly winds brought a significant sediment flux from southeast to northwest along the JC, with a large magnitude. The typhoon that did not land and moved northward in the Yellow Sea brought a relatively smaller sediment flux from north to south. Additionally, typhoons caused intense stripe-like erosion and deposition in the sand ridges and deep channels, with maximum seabed thickness variations of up to ±0.3 m. The strengthening of the typhoon wind fields led to more significant sediment flux and morphological changes in the seabed.

(3) During the impact of Typhoon Winnie within the spring tide cycle, its influence on sediment dynamics was greater than that of other landfall typhoons. Through numerical simulation experiments, it was found that if Typhoon Lekima affected the JC during spring tides, the net sediment flux within the tidal cycle could increase by 80% to 100% compared to the neap tide scenario. In the absence of tidal effects, the net sediment flux caused by the typhoon would be only 30–50% of that in the neap tide scenario. Furthermore, in the spring tide scenario, the impact of the typhoon on

coastal geomorphological changes would be significantly greater than that under the neap tide or no-tide conditions.

This study demonstrates that not only do the track and intensity of typhoons affect seabed sediment dynamics, but tidal effects also play a fundamental and crucial role in this process, especially under the spring tide conditions where tidal effects significantly amplify the impact of typhoons on sediment dynamics. While these findings are site-specific, tthey contribute to a better understanding of the tidal effects and differences of TCs under different scenarios and provide a theoretical basis for coastal management and disaster prevention and mitigation.

**Author Contributions:** Conceptualization, G.Y., C.L., C.Z. and J.Z.; Methodology, G.Y. and C.L.; Software, C.W., G.Y. and C.L.; Validation, C.W.; Formal Analysis, C.W.; Investigation, C.L.; Resources, X.M.; Data Curation, C.W.; Writing—Original Draft Preparation, C.W.; Writing—Review and Editing, J.Z. and C.Z.; Visualization, C.W.; Supervision, C.Z.; Project Administration, C.Z.; Funding Acquisition, C.Z. All authors have read and agreed to the published version of the manuscript.

**Funding:** This work was sponsored by the key project of the National Natural Science Foundation (42130405) and the Innovative and Entrepreneurial Talent Program of Jiangsu Province (R2020SC04).

**Institutional Review Board Statement:** Not applicable.

**Informed Consent Statement:** Not applicable.

**Data Availability Statement:** The data presented in this study are available on request from the corresponding author.

**Conflicts of Interest:** The authors declare no conflicts of interest.

## Appendix A

To strengthen the robustness of the model's prediction performance, we collected field observation data in 2012 during Typhoon Damrey for model calibration. The results show that the simulation results are highly consistent with the observation results, which indicates that the coupled model has high credibility in predicting dynamic processes during TCs.

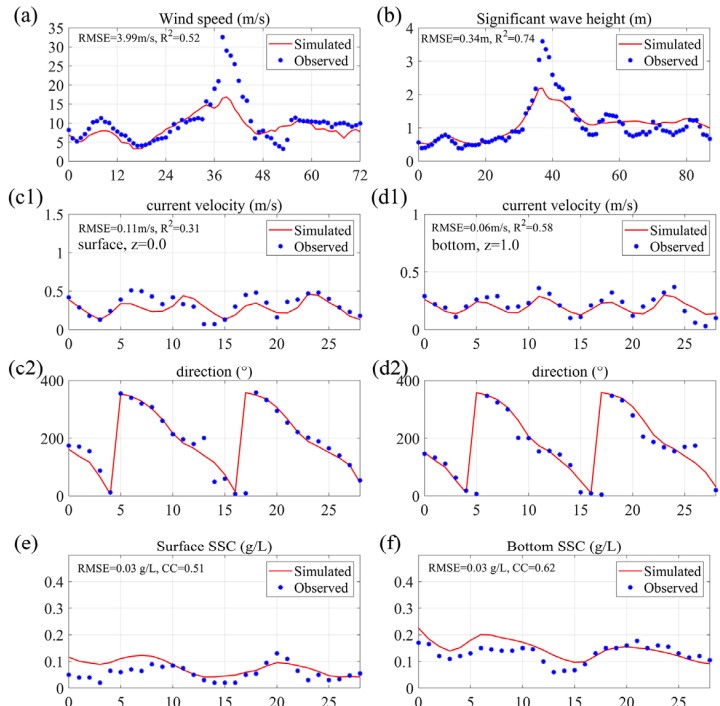

**Figure A1.** *Cont.*

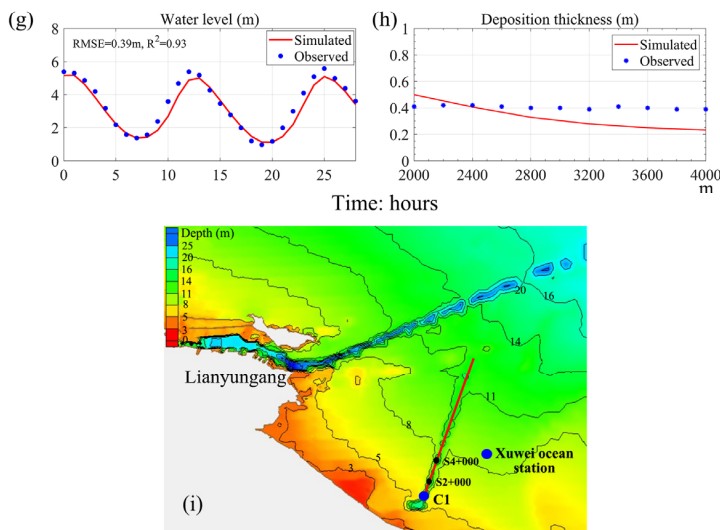

**Figure A1.** Comparisons between the observed and simulated (**a**) wind speed and (**b**) significant wave height during Typhoon Damrey (2012) at Xuwei Ocean Station. Comparisons between observed and simulated (**c1**,**d1**) surface and bottom current velocity, (**c2**,**d2**) direction, (**e**,**f**) surface and bottom SSC, and (**g**) water levels at point C1 during August 2012. (**h**) Validation of deposition thickness at sections S2–S4 in the Xuwei approach channel after the passage of Typhoon Damrey. (**i**) Bathymetry map and the location of field stations.

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
