# Peer review of "Response of Sediment Dynamics to Tropical Cyclones under Various Scenarios in the Jiangsu Coast"

_jmse, doi:10.3390/jmse12071053_

Round 1

Reviewer 1 Report

Comments and Suggestions for Authors

This work is very interesting and takes into account the influence of tropical cyclones for the sedimentary dynamics. Here, dynamic models were applied to assess the sedimentary dynamics in the function of typhoons occurred in China. I have just two comments for the manuscript. First, in the abstract it was reported that modelling was performed under various scenarios. Maybe it is better to define briefly in the abstract what are these scenarios and then you can explain better in material and methods. Second, you must define the tidal conditions existing in your study areas (like macrotidal, semidiurnal, amplitude, etc.). This is important to discuss the increase of water level due to the occurrence of typhoons during spring tides.

Other minor comments are highlighted in the attached pdf file.

Kind regards.

Reviewer 2 Report

Comments and Suggestions for Authors

This paper is very interesting and provides a number of great results. However, the English is poor, with a lot of typos. It is strongly required to make some corrections regarding grammar. The details of suggestions should refer to the file attached

Comments on the Quality of English Language

Reviewer 3 Report

Comments and Suggestions for Authors

The manuscript entitled “Response of Sedimentary Dynamics to Tropical Cyclones 2 under Various Scenarios in Jiangsu Coast” is of a very interesting topic. The authors developed a 3D wave-current-sediment numerical model for Jiangsu Coast based on the Finite Volume Coastal Ocean Model (FVCOM) to investigate sediment dynamic response to tropical cyclones in various scenarios. It is of great importance that the authors in an attempt to strengthen the robustness of the model prediction performance they collected field observation data in 2012 during Typhoon Damrey for model calibration. The topic of the manuscript is absolutely within the scope of the Journal of Marine Science and Engineering.

The manuscript is well written and well structured. I enjoyed reading it! I am not a native speaker but I have to admit that the manuscript reads quite well!

Here are some comments and suggestions of minor importance which could improve the final version of the paper:

I was just wondering if the application of such a model can provide information about palaeo-storm surge events. This could be of great importance for those who deal with the identification of such events taking into consideration the stratigraphy of the coastal sediments from core sampling. If the application of the model can help in this direction it would be good to be mentioned in the “Introduction” section.

In my opinion the “Introduction” section should have a brief literature review of similar methodologies or approaches and their limitations.

A Figure depicting the study area, its morphology (both on land and offshore) as well as its location in China would be more than appreciated by the readers of the final version of the paper.

Congratulation to the authors,

E. Karymbalis

Reviewer 4 Report

Comments and Suggestions for Authors

Introduction - Highlights the potential impact of tropical cyclones along the east coast of China and makes concise reference to a number of key studies that highlight the importance of TCs to sediment dynamics. Also notes the lack of observational data available for the selected study area.

Regional Setting and study Domain - Provides a brief description of the study area and its climate with additional detail provided related to one significant TC. The section would be improved by describing the tidal ranges and providing a map that illustrates the details noted in the text (e.g. could note the 4 coastal zones and the variations in tidal range)

Methodology - The text clearly outlines how the models were developed and calibrated. The sources of data and the parameters used are clearly outlined. The methodology appears to be appropriate.

Results - The results from the modelling are clearly described and supported by appropriate diagrams/maps. The impact of the different TC tracks is noted as well as the influence of various tidal conditions. General models of sediment erosion and deposition are noted .

Discussion - Reference is made to one study where observtional data is discussed which supports the findings from the models. There is also a detailed discussion of the impact of tides on the sediment dynamics. The discussion could be improved if reference was made to studies from elsewhere around the world which discuss the morphological impacts of Hurricanes/Typhons/Cyclones. Studies with observational data which support the model findings or which produced contrasting results should be noted. There also appears to be limited discussion of shoreline changes predicted by the models.

Conclusion - Provides an appropriate summary of the paper.

Minor Changes

Line 13 - Change to ....highly impacted by tropical cyclones (TCs)

Line 14 - remove ...available on-site...

Line 20 - What is SSC?

Lines 35, 36 and 38 - remove , before and. Check the rest of the text and remove any , before and.

Line 97 - Except should not have a capital E

The text uses meter for the unit of length rather than metre (so the American rather than other English speaking countries spelling). Given the measurement was originally defined in France and was spelt as metre I would suggest that meter is replaced with metre throughout the text.

Comments on the Quality of English Language

Quality of English is generally very good. Use of the Oxford comma (e.g. before and) should be removed. Spelling of metric unit of length should reflect the origin of the term rather than adopt American English.

Reviewer 5 Report

Comments and Suggestions for Authors

It is a very important and interesting paper, analyzing the impact of typhoons in a segment of the Eastern coast of China. The general analysis is well done, the methodology is strong and well presented, the references are necessary and well used, the conclusions are substantial. Some questions are yet not clear, for example, the reason for not discussing the impacts of the typhoons on the coastline (mainly in the seabed). It should be clear in the presentation what are the objectives of the paper. We highlighted some questions  below, also informing that there is the need for some more illustrations and more evidence about the system of classifications of typhoons in China. Nevertheless, in a general point of view, the data merit to be known by the scientific Community.

Line 41 – There is no future global warming, global warming is already a fact. The authors should suppress the word “future”

Line 46 – The text should inform the year of occurrence of these typhons

Line 49 – the same Thing as the precedent line (the year of the typhon)

Line 68 – It is not clear what are the typhoons that impact the coastal áreas. What is the difference between them? The others are not in the coastal áreas as well?

Line 84 – It lacks a localization map of the study área

Line 272 – It is not clear the date ’14 days ago’

Line 378 – how was the erosion in the coastline?

Line 533 – The conclusion should talk about seabed sediment dynamics, and not coastal dynamics, because the dynamics of the coastal sectors were not analyzed.

Round 2

Reviewer 2 Report

Comments and Suggestions for Authors

I agree with the revised version